# Influence of the Demographic, Social, and Environmental Factors on the COVID-19 Pandemic—Analysis of the Local Variations Using Geographically Weighted Regression

**DOI:** 10.3390/ijerph191911881

**Published:** 2022-09-20

**Authors:** Krzysztof Rząsa, Mateusz Ciski

**Affiliations:** Faculty of Geoengineering, Institute of Spatial Management and Geography, Department of Land Management and Geographic Information Systems, University of Warmia and Mazury in Olsztyn, 10-720 Olsztyn, Poland

**Keywords:** COVID-19, SARS-CoV-2, pandemic, geographically weighted regression, GWR, geographic information system, GIS

## Abstract

As the COVID-19 pandemic continues, an increasing number of different research studies focusing on various aspects of the pandemic are emerging. Most of the studies focus on the medical aspects of the pandemic, as well as on the impact of COVID-19 on various areas of life; less emphasis is put on analyzing the influence of socio-environmental factors on the spread of the pandemic. In this paper, using the geographically weighted regression method, the extent to which demographic, social, and environmental factors explain the number of cases of SARS-CoV-2 is explored. The research was performed for the case-study area of Poland, considering the administrative division of the country into counties. The results showed that the demographic factors best explained the number of cases of SARS-CoV-2; the social factors explained it to a medium degree; and the environmental factors explained it to the lowest degree. Urban population and the associated higher amount and intensity of human contact are the most influential factors in the development of the COVID-19 pandemic. The analysis of the factors related to the areas burdened by social problems resulting primarily from the economic exclusion revealed that poverty-burdened areas are highly vulnerable to the development of the COVID-19 pandemic. Using maps of the local R^2^ it was possible to visualize how the relationships between the explanatory variables (for this research—demographic, social, and environmental factors) and the dependent variable (number of cases of SARS-CoV-2) vary across the study area. Through the GWR method, counties were identified as particularly vulnerable to the pandemic because of the problem of economic exclusion. Considering that the COVID-19 pandemic is still ongoing, the results obtained may be useful for local authorities in developing strategies to counter the pandemic.

## 1. Introduction

In December 2019, the Chinese government reported the first cases of the SARS-CoV-2 infection in Wuhan, the capital of Hubei province [1,2]. The epidemic quickly spread to all the provinces in China. More cases began to appear in countries around the world. On 11 March 2020, the World Health Organization declared this coronavirus outbreak a pandemic [3]. The disease the virus causes was named COVID-19; the COVID-19 pandemic has changed the world today, becoming one of the major health challenges of the 21st century. Having influenced many areas of life, it has changed the lives of many people around the world. Almost two years after its beginning, it also became the subject of scientific research.

Most of the studies that have been conducted and published focus on the medical aspects of the COVID-19 pandemic: the effects on human health [4,5,6,7,8,9,10,11,12], as well as the process and impact of vaccination [13,14,15,16,17,18,19]. The researchers analyzed the impact of the COVID-19 pandemic on lifestyle behaviors and mental health; the pandemic is negatively impacting self-perceived physical and mental health, particularly among people living with non-communicable diseases [20,21,22,23,24]. Public health initiatives are needed to address healthy lifestyle behaviors during and after the COVID-19 pandemic. A possible mitigation strategy for improving mental health includes taking suitable amounts of daily physical activity and sleeping well. The COVID-19 outbreak has reduced people’s aggressiveness, probably by making people realize the fragility and preciousness of life [20].

Many studies have attempted to determine the impact of the COVID-19 pandemic on various areas of life. The COVID-19 pandemic, through a series of introduced restrictions and limitations, has changed the way people use public space; this has become particularly apparent in the case of green spaces [25,26,27,28,29,30,31,32]. Urban open space played a positive role in allowing people to stay connected with their neighbors, to feel reassured, and to maintain or increase their physical activity levels during the pandemic. The reason for visiting green spaces much more often during lockdown was to take care of one’s mental well-being, as well as for recreation. The perception of public spaces has changed, and their accessibility and uses have changed [33,34,35,36]. The COVID-19 pandemic has influenced the functioning and planning of urban areas—with half of the globe forced into lockdown, urban planners needed a new approach to the use of space in many aspects of urban activity and population mobility [37,38,39,40,41,42,43,44,45]. Employment has a significant impact on mental health, especially in the context of a global pandemic. The workplace is therefore an important target at which efforts should be directed to manage the mental health issues associated with the COVID-19 pandemic [46,47,48]. The COVID-19 pandemic has unmasked the problems related to uncertainty about the economic and employment aspects; this is critical for policy design and financial strategy and planning [49,50,51,52,53,54,55,56]. The real estate market also transformed during the COVID-19 pandemic; many households are reconsidering their housing needs as their homes have become substitutes for offices, schools, restaurants, and recreational facilities [57]. Changes in energy production and a reduction in fossil fuel consumption before and during the COVID-19 pandemic, as well as the reduced traffic and industrial activity in 2020, can explain the lower tropospheric NO_2_ emission concentration [58].

The research related to the COVID-19 pandemic has used GIS tools in addition to a range of statistical tools [59,60,61,62,63,64,65,66]. GIS technology is very useful for aspects of research related to the analysis of COVID-19, the spatial spread of the epidemic, and other related aspects of the pandemic. Monitoring new cases using GIS spatial analysis can be very useful for controlling the course of the pandemic.

While the impact of the COVID-19 pandemic on various aspects related to human life has been analyzed by researchers around the world, less emphasis was put on analyzing the influence of the various socio-environmental factors on the spread of the pandemic. The studies show that housing quality, living conditions, demographic status, and occupation were strong factors influencing the spread of the pandemic, as well as the mortality rate of the COVID-19 cases [67]. More populous and densely populated places have a higher risk of transmission of COVID-19, especially with the Delta variant as the dominant circulating strain. Therefore, extra control measures should be instituted in highly populated areas to control the spread [68]. The temperature variation and humidity may also be important factors affecting COVID-19 mortality [69]; mean temperature has a positive linear relationship with the number of COVID-19 cases when the temperature is below 3 °C [70].

The purpose of this article is to analyze the influence of the socio-environmental factors on the spread of the COVID-19 pandemic. The extent to which various demographic, social, and environmental factors explain the number of cases of SARS-CoV-2 is explored. Poland was selected as a case-study area. The study’s introduction is a description of the variables analyzed and a spatial autocorrelation test. The obtained database contained information on various demographic, social, and environmental factors, as well as the number of cases of SARS-CoV-2. The influence of the collected factors on the expansion of the COVID-19 pandemic in Poland was examined using the geographically weighted regression method (GWR), with ArcGIS Pro 2.9 software (by Esri, Redlands, CA, USA). GIS tools are widely used in various types of spatial analysis [71,72,73].

The stated research question was to investigate the influence of non-medical factors (demographic, social, and environmental) on the COVID-19 pandemic in Poland. Therefore, the following research hypothesis was established: using advanced statistical models, it is possible to indicate the influence of non-medical factors on the development of a pandemic. An additional hypothesis of the article is the following: with the local variances of the results obtained with the GWR method, it is possible to indicate anomalies in relation to the overall, national results, which can be a very useful basis for taking additional measures by local authorities in the further fight against the COVID-19 pandemic.

The results of the conducted research expand the relatively small knowledge in this area and can be used in further efforts to prevent the spread of the COVID-19 pandemic, which is still active worldwide.

## 2. Materials and Methods

### 2.1. Study Area

Research on the influence of various demographic, social, and environmental factors, as well as the number of cases of SARS-CoV-2, was carried out for the area of Poland. As the level of accuracy, the counties were chosen—this is the second level of the administrative division of the country (the first is the voivodeships, the third is the municipalities). Poland is divided into 380 counties, with an average area of approximately 822 square kilometers. Among the counties, there are 66 cities with county rights—these are municipalities with city status, executing county duties. Swietochlowice city county is the smallest county (13 square kilometers), and Bialystok county is the largest (2975 square kilometers).

The chosen level of accuracy of the study will allow a thorough examination of the phenomenon (due to the relatively small area of the analyzed polygons) while maintaining a higher administrative level, which brings greater data availability. In addition, the GWR method chosen for the study requires at least 20 features to compute the results and achieves the best results with larger datasets [74]. The number of counties in Poland meets the requirements of the GWR method. Figure 1 presents the study areas—the county and voivodeship boundaries in Poland. 

### 2.2. Data Source and Processing

The basis of the study was a database containing the number of cases of SARS-CoV-2 in the Polish counties and a variety of demographic, social, and environmental factors. The chosen spatial extent of the analysis, i.e., the counties of Poland, is the most spatially accurate level for which the data on the number of cases of SARS-CoV-2 in Poland have been published. The number of cases of SARS-CoV-2 in the counties in Poland was obtained from data collected from reports provided by the University of Warsaw, the Voivodeship Sanitary and Epidemiological Stations, and the County Sanitary and Epidemiological Stations and from materials obtained from requests for access to public information, as well as those collected from reports provided by the Ministry of Health; the database was published by Michal Rogalski and Konrad Kalemba [75,76,77]. This is a reliable source of data, used by many scientific studies examining the issue of the COVID-19 pandemic in Poland [14,78,79,80,81,82,83,84]. The following, in Figure 2, presents the spatial distribution of the summary of the cases of SARS-CoV-2 in the counties of Poland, as of the end of 2021.

In order to examine the possible influences of the demographic, social, and environmental factors on the COVID-19 pandemic in Poland, it was necessary to determine the state of these factors before the outbreak of the pandemic (creating a depiction of the demographic, social, and environmental state of Poland as of December 2019). This is what dictated the choice of the time range of the variables. The first cases of SARS-CoV-2 in Poland were registered in March 2020; to study the phenomenon as widely as possible, case data from two consecutive years (cases from the beginning of the pandemic in 2020 to 31 December 2021) were used.

All the data concerning the selected factors came directly from Statistics Poland, the central office of government administration in Poland, which collects and shares statistical information on the national level. This data source is reliable and provided by a public institution; it is used in various scientific studies [85].

The research indicating the state of the study area in terms of demographic, social, and environmental conditions often relies on a very broad set of indicators. It is related to the purpose of the research, the thematic and spatial scope, and the level of detail. In the research conducted within this article, the biggest limitation in the selection of indicators was the level of spatial detail of the analyses (the research was conducted for the second level of Poland’s administrative division). Most of the data published in Statistics Poland are for the national or voivodeship level (the first level of administrative division). The characteristics of the study required greater detail in the data, which limited the choice of specific indicators the most.

Considering the above limitations, based on the analysis of the literature, a number of demographic, social, and environmental indicators were selected for the research conducted in this article [86,87,88,89,90,91,92,93,94,95]. The following in Table 1 lists the selected factors (variables), broken down by thematic sections. In order to make the content and results more readable, each variable was attributed with a symbol, consisting of the first letter of the section’s name and the subsequent ordinal number (e.g., the variable “Households receiving community social assistance” is located in the “Social” section and is the fourth variable; so, it received the symbol “S4”). Replacing the variable name with a symbol will increase the readability of the content; the symbolic designation is retained in the rest of the article. The full database used for the research in this article can be found in the Appendix A.

The data on the structure of the population in the counties were gathered in Statistics Poland from the National Population Censuses; the data include changes resulting from births and deaths, as well as the migration of the population (for permanent residence and for temporary stay) and changes caused by administrative changes. Data on the place of residence of the population were collected on the basis of the PESEL register. The “Demographic” section is a detailed cross-section through the population, distinguishing place of residence (urban–rural) and broken down into four age classes.

The “Social” section can be divided into two thematic groups: variables S1–S3 describe the level of healthcare and medical services, and variables S4–S7 identify areas burdened by social problems resulting primarily from the economic exclusion of the population. The state of health care, represented by the number of beds in hospitals, the number of doctors, and the number of nurses, is obtained from the Center for Health Information Systems, which is the institution responsible for the operation of the register of the entities performing medical activities in Poland. The areas burdened by the economic exclusion of the population were determined on the basis of the indicators of the living conditions of the population: families benefiting from community social assistance and families with assistance on the basis of poverty, as well as social welfare: families receiving family benefits for children and payments from the government child-raising benefit 500+ program.

The “Environmental” section can also be divided into two thematic groups. The variables E1 and E2 describe the state of environmental pollution through the magnitude of the release of dust and gaseous pollutants into the atmosphere in an organized (through stationary point sources) or unorganized manner (from dumps, landfills, during reloading of loose or volatile substances, through roof and window ventilation, due to forest fires, etc.). These are the emissions whose concentration exceeds the average content of these substances in the clean air, negatively impacting human health and the condition and quality of the environment. Variables E3 and E4 describe the country’s forest cover, as well as the areas covered with vegetation, located in the villages of dense buildings or cities and used for aesthetic, recreational, therapeutic, or shielding purposes, in particular parks, lawns, promenades, boulevards, botanical and zoological gardens, children’s playgrounds, historic gardens, cemeteries, or other green areas located in the built-up areas.

Tabular data containing the number of cases of SARS-CoV-2 in the counties of Poland and a variety of demographic, social, and environmental factors were merged and combined with the polygon data representing the administrative boundaries of the counties of Poland. The polygon data were obtained from the Polish National Register of Boundaries (NRB) [96]. The NRB is an official Polish spatial database, which forms the foundation for other spatial information systems and shares data concerning the administrative units of the country. The NRB covers the area of the whole country and contains information on the boundaries and basic attributes of the three-tier administrative division of the country (i.e., municipalities, counties, and voivodeships). Thus, the obtained geodatabase was used for further research. The following, in Figure 3, presents the spatial distribution of the used variables in the counties of Poland.

### 2.3. Research Method

Prior to the study, the authors decided to establish the null hypothesis—there is no statistical significance in all of the analyzed variables (in the case of this research—in the various demographic, social, and environmental factors, as well as in the number of cases of SARS-CoV-2 in the counties of Poland). The purpose of the null hypothesis is to test for statistical significance in the data and to verify that the results obtained are not the result of random chance. In order to assess the probability of the null hypothesis being true or false, a spatial autocorrelation analysis (Global Moran’s I) was carried out using the ArcGIS Pro 2.9 software (by Esri, Redlands, CA, USA). Spatial autocorrelation is the presence of systematic spatial variation in a variable; positive spatial autocorrelation is the tendency for areas or places close together to have similar values. The most common method for measuring spatial autocorrelation is to calculate the Moran’s I index [97,98,99,100,101]. The hypothesis test is as follows: if the Moran’s I = 0, there is no spatial autocorrelation; if the Moran’s I > 0, spatial autocorrelation exists, with positive and negative values indicating positive and negative autocorrelation.

Once the null hypothesis is rejected, it is possible to proceed to an examination of the influence of the selected variables on the number of cases of SARS-CoV-2 in the Polish counties using geographically weighted regression. Geographically weighted regression (GWR) is a method of analyzing spatial data based on regression, developed by adding local spatial weights [102,103,104,105,106]. GWR models considerably improve the modeling fit by capturing spatial heterogeneity, which is not factored into other regression models [101,106]. This model fully aligns with the first law of geography. The GWR model can be expressed as:(1)$$y_{i}=\beta_{0}\left(u_{i},v_{i}\right)+\overset{p}{\underset{k}{\sum }}\beta_{k}\left(u_{i},v_{i}\right)x_{ik}+\varepsilon_{i}$$
where
$y_{i}$—dependent variable at location *i*$x_{ik}$—explanatory variable at location *i*$\left(u_{i},v_{i}\right)$—coordinate for location *i*$\beta_{0}\left(u_{i},v_{i}\right)$—intercept location *i*$\beta_{k}\left(u_{i},v_{i}\right)$—coefficient for explanatory variable *k* at location *i*$\varepsilon_{i}$—residual location *i*

The geographically weighted regression method has found application in studies regarding various infectious diseases: COVID-19 [107,108,109], AIDS [110], the Zika virus [111,112], tuberculosis [113], malaria [114], and others [115,116]. The GWR method can be used to estimate the effects of explanatory variables (in the case of this study, a number of social, demographic, and environmental factors) on the dependent variable (number of cases of SARS-CoV-2 in Polish counties) and also to identify the counties in which the influence of the variables differs and to explore and interpret the spatial non-stationarity. The ArcGIS Pro 2.9 software (by Esri, Redlands, CA, USA) was used to perform the GWR analysis.

In order to estimate the influence of the explanatory variables on the dependent variable, the R^2^ parameter was used. R^2^ (or R-squared) is the proportion of the dependent variable variance accounted for by the regression model; it is a measure of the goodness of fit and quantifies the performance of a local GWR model. R^2^ is called the coefficient of determination; its value varies from 0.0 to 1.0, and a higher value means that the explanatory variable explains the dependent variable better [99,117]. 

In order to evaluate whether or not the GWR model is biased by other factors, the spatial autocorrelation (global Moran’s I) tool was performed on the regression residuals. An unbiased model has residuals that are randomly scattered [106,118,119]. The explanatory variable explains the dependent variable well only after two requirements are met: a high value of the coefficient of determination (e.g., R^2^ > 0.8) as well as the weak or insignificant level of spatial autocorrelation in the residuals [106].

## 3. Results

### 3.1. Spatial Autocorrelation Analysis

In order to evaluate the likelihood that the null hypothesis is true or false, spatial autocorrelation (global Moran’s I) analysis was performed using ArcGIS Pro 2.9 software (by Esri, Redlands, CA, USA). The spatial autocorrelation (global Moran’s I) tool measures spatial autocorrelation on the basis of feature locations and feature values; it assesses whether the expressed pattern is clustered, dispersed, or random. Below (Figure 4) are the spatial autocorrelation (global Moran’s I) reports of the analyzed variables.

A positive z-score indicates a tendency to form clusters, and it has a negative aspect—a tendency for the data to be dispersed [100,101]. The following, in Table 2, summarizes the results of the spatial autocorrelation, with the *p*-value, the z-score, an indication of the spatial pattern, and the confidence level for all the analyzed variables.

The obtained indicators made it possible to reject the original null hypothesis. Sixteen out of eighteen variables are statistically significant at the 0.01 confidence level, and two variables are statistically significant at the 0.05 confidence level. The positive z-scores indicate the variables that are mostly spatially clustered, and the data are characterized by spatial heterogeneity. This allows the research to continue; the next step was to assess the influence of the selected variables on the number of cases of SARS-CoV-2 in Polish counties using geographically weighted regression. 

### 3.2. Influence of the Selected Variables on the Number of Cases of SARS-CoV-2

The analysis was performed using ArcGIS Pro 2.9 software (by Esri, Redlands, CA, USA). The polygon data representing the county boundaries in Poland were used for the study. The tabular data containing the selected variables were combined with the polygon data representing the polish counties, using the ‘Joins and Relates’ tool. Using the collected explanatory variables, the GWR analysis of the influence of the variables on the number of cases of SARS-CoV-2 in the Polish counties was carried out. The indicator values for the selected variables, as well as a statistical description of the variables, are shown in Table 3 below. 

Among the analyzed variables, variable D1, the “total population,” is characterized by the highest R^2^ value. All variables in the “Demographic” section exhibit a very high R^2^ value, with the exception of variable D3 “rural population”. These results indicate that the spread of the COVID-19 pandemic took place primarily in cities; it can be closely associated with densely populated areas. The differences between the R^2^ values for the four age divisions (variables D4–D7) are marginal but indicate that young people (in the 16–25 age range) have a slightly smaller influence on the development of the COVID-19 pandemic in Poland.

The variables in the “Social” section also had a strong influence on the dependent variable: for variables S1, S4, S5, S6, and S7, the value was R^2^ > 0.80 (the highest value was obtained by the variable S7 “Benefit payments from the 500+ program”), while with variables S2 and S3 a smaller impact was observed, at R^2^ = 0.73 and R^2^ = 0.66, respectively. Variables S1–S3 determine the level of health care in the studied counties, while variables S4–S7 reflect the level of social welfare and thus identify the areas burdened by social problems resulting primarily from poverty. The areas most affected by these social problems appear to be the most susceptible to SARS-CoV-2.

The variables in the “Environmental” section did not appear to significantly influence the number of cases of SARS-CoV-2; only variable E4 has a significant R^2^ = 0.83.

### 3.3. Local R^2^ Estimates

GWR allows for the exploration of spatially varying relationships. In order to estimate the spatial distribution of different variables, the local R^2^ estimates are presented below (Figure 5, Figure 6 and Figure 7). This is a way to visualize how the relationships between the explanatory variables and the dependent variable vary across the study area. Mapping the local R^2^ estimates may provide clues about important variables that may be missing from the regression model; for this study, the focus was on whether or not the number of cases of SARS-CoV-2 in Polish counties was influenced by factors other than the examined variable.

Selecting the appropriate data classification and symbolization method is an extremely important part of the cartographic process [120]. The class selection was carried out using the Jenks natural breaks method. This method is one of the most popular data classification methods [85,121,122]; it is used to reduce the variance within classes and to maximize the variance between classes. The comparison of the content of the maps is only possible if equal classification (division into classes) is used [123]. To enable the comparison of the local R^2^ distribution for all the explanatory variables in all three sections, the classification of the maps was standardized. This way, each class on each map corresponds to the same values; this allowed us to draw valid conclusions without any spatial-classification bias.

The boundaries of the voivodeships were overlaid on the maps; the voivodeships do not have full autonomy, but they do have some administrative independence, primarily in terms of decision making and the disposition of resources. Such a procedure allowed for additional analysis of the impact of voivodeship government actions on the progress of the COVID-19 pandemic in Poland. The following, in Figure 5, presents the local R^2^ values for the “Demographic” section.

The use of local R^2^ prediction maps revealed a significant variance in the demographic variables analyzed in the study area. The majority of the counties in the Masovian voivodeship have the highest local R^2^ values for all the studied variables. For the majority of variables, the counties in the southwestern part of Poland indicate a very large deviation from the R^2^ value for the country. By overlaying a layer of the administrative boundaries of the voivodeships, in the case of the demographic data, a significant effect of the location of the counties in a given voivodeship on the local variance of the R^2^ parameter can be observed (this is evident when the administrative boundary of a voivodeship overlaps with counties belonging to two different symbolization classes). Within the boundaries of a single voivodeship, the counties are often in up to two (and sometimes even one) symbolization classes.

In the case of the explanatory variable D1 (total population), the counties located in the West Pomeranian, Greater Poland, Łódź, and Masovian voivodeships appear to explain the explanatory variable the best (the local R^2^ values are in the fifth class, which is the highest); on the other hand, the counties located in southwestern Poland explain the explanatory variable the least effectively. The spatial distribution of the local R^2^ for the D2 variable almost matches that for the D1 variable. The map of the D3 variable shows an unusually large number of counties for which the local R^2^ values are negative or close to zero. This indicates a low level of explanation of the dependent variable, which is confirmed by the overall R^2^ for this explanatory variable (R^2^ = 0.37). Examining the age structure of the population (four variables D4–D7), it is noticeable how the results obtained are not significantly different from the results obtained for variable D1. For the D6 variable, which best explains the dependent variable (R^2^ = 0.99), no county is characterized with a negative local R^2^. 

The following, in Figure 6, shows the local R^2^ values for the “Social” section. The bottom range of the lowest class was slightly modified to account for the decreasing local R^2^ value.

In the “Social” section, two thematic groups of explanatory variables can be distinguished: variables S1–S3 describe the level of healthcare and medical services, and variables S4–S7 identify the areas burdened by social problems resulting primarily from the economic exclusion of the population. For both thematic groups, the local R^2^ maps show far greater heterogeneity. Most of the counties in the Masovian voivodeship for almost all the explanatory variables seem to best explain the dependent variable. In addition, again the regions of southwestern Poland and the eastern counties of the West Pomeranian voivodeship explain the dependent variable the least.

In the case of the first thematic group, one can see the preservation of spatial relationships between the local R^2^ values. The areas indicating a lower value of local R^2^ appear to be almost the same across all three maps. It is most noticeable in the following areas: the eastern part of the West Pomeranian voivodeship and the southwestern part of the country, as well as almost all of the Holy Cross and Masovian voivodeships. The described counties might belong to different symbolization classes, but their mutual relationship is preserved. A similar pattern can also be observed in similar areas for the second thematic group, especially in the counties of the Masovian and West Pomeranian voivodeships.

As in the case of the “Demographic” section, one can see a large impact of the location of the counties in a given voivodeship on the local variance of the R^2^ parameter. For instance, this is clearly seen in the case of explanatory variables S6 and S7.

The following, in Figure 7, illustrates the local R^2^ values for the variables in the “Environmental” section. The bottom range of the lowest class was again slightly modified to account for the further decreasing local R^2^ value.

The maps of the local R^2^ of the “Environmental” section demonstrate the highest heterogeneity overall by far, higher than the other two sections. In addition, compared to the other sections, more counties show negative local R^2^ values. The highest local R^2^ values were again obtained in the counties of the Masovian voivodeship. Negative values of local R^2^ appear throughout the country but again dominate in the southwestern part of the country.

The map of the local R^2^ of the E3 variable “forest cover” strongly deviates from the rest of the results. In this particular case, the authors were forced to use a different classification of the data since the local R^2^ values are in the range of 0.013–0.026. Applying the same classification as for the other local R^2^ maps would place the entire case study area in a single class of 0–0.3. This situation is due to the extremely low R^2^ value for the entire variable, which is 0.02. 

### 3.4. Residual Spatial Autocorrelation

In order to evaluate the performance of the GWR model, the existence of residual spatial autocorrelation was analyzed. The spatial autocorrelation (global Moran’s I) test was again performed using the ArcGIS Pro 2.9 software (by Esri, Redlands, CA, USA) to measure the residual spatial autocorrelation. The results are shown in Table 4 below.

Given the z-scores, the pattern of residuals does not appear to be significantly different than the random. Only for variables D4, D6, D7, and S7 were values indicating a weak dispersed pattern recorded (z-score between −1.96 and −1.65). The results indicate the random nature of the residuals, confirming the validity of using the GWR model.

## 4. Discussion

In the available databases of academic articles, the influence of the COVID-19 pandemic on various aspects related to human life has been quite thoroughly analyzed. Less emphasis was put on analyzing the influence of the various socio-environmental factors on the spread of the pandemic. The results of this research expanded the relatively small knowledge on the impact of the various factors on the progress and course of the COVID-19 pandemic. 

The conducted research confirmed the research hypotheses. The established goal of the research, which was to assess the influence of various demographic, social, and environmental factors on the number of cases of SARS-CoV-2, was implemented using the geographical weighted regression (GWR) method. R^2^ parameter values, or the coefficients of determination, were compiled; the values vary from 0.0 to 1.0, and a higher value means that the explanatory variable explains the dependent variable better. For this research, this means that the higher the R^2^ value for a demographic, social, or environmental variable, the better that variable explains the number of cases of SARS-CoV-2. 

The highest values of the R^2^ parameter were obtained for the variables in the “Demographic” section; the values for most of these variables ranged from 0.97 to 0.99. This confirms that population and the associated higher amount and intensity of human contact are the most influential factors in the development of the COVID-19 pandemic. The low R^2^ = 0.37 result for variable D3 “rural population” also confirms this point. The rural areas are characterized by a smaller population and a greater degree of dispersion, which implies fewer person-to-person interactions. The rural areas also tend to have reduced population mobility, which may restrict the spread of the virus. 

The “Social” section can be subdivided into two thematic groups: the S1–S3 variables describe the level of health care and medical services, and the S4–S7 variables identify the areas burdened by the economic exclusion of the population. The R^2^ parameter for the variables in the “Social” section obtains values at an average level, in the range of 0.66–0.97. This is still a high degree of explanation of the dependent variable but clearly lower than the variables in the “Demographic” section. The highest R^2^ value in this section was recorded for variable S1—number of beds in general hospitals. This may be related to the fact of the increased number of healthcare facilities in urban areas, where a higher number of cases of SARS-CoV-2 virus infection have occurred. Negative R^2^ values were obtained in the counties of southwestern Poland: Bolesławiec county, Zgorzelec county, and Żagań county. The R^2^ values for variables S2 and S3, although thematically linked to variable S1, are lower, at 0.73 and 0.66, respectively. Accordingly, the number of medical personnel explains the dependent variable to a lesser extent. The areas indicating a negative value of local R^2^ appear across all the maps. The thematic group of the variables S3–S7, indicating the areas burdened by social problems resulting primarily from the economic exclusion of the population, is characterized by a high level of the R^2^ parameter. This strongly implies that the poverty-burdened areas are also highly vulnerable to the development of the COVID-19 pandemic.

The lowest R^2^ values (in the range of 0.02 to 0.83) for the “Environmental” section variables indicate a lower degree of explanation for the SARS-CoV-2 cases. The highest R^2^ value in this section was registered for variable E4, i.e., the share of parks, greens, and neighborhood green areas. This is another explanatory variable that is strongly associated with urban areas, which may explain the obtained result. The variables associated with the environmental pollution issues (E1 and E2) moderately explain the number of cases of SARS-CoV-2 infection. The R^2^ for these variables is 0.72 and 0.70, respectively. The lowest value of the R^2^ parameter was recorded for variable S3, i.e., forest cover—the obtained level of R^2^ = 0.02 indicates a complete lack of association of this variable with the dependent variable.

Thus, the results showed that the demographic factors best explained the number of cases of SARS-CoV-2, the social factors explained it to an average degree, and the environmental factors explained it to the lowest degree.

## 5. Conclusions

The application of the GWR method is not common in COVID-19 pandemic studies. In addition, most the research of this type is carried out at the country level. In the research conducted in the article, the authors developed maps of local R^2^ at a high level of spatial detail (research at the level of Poland’s second administrative division), allowing for more detailed conclusions and the formulation of more specific guidelines for local authorities.

The GWR method allowed for the exploration of spatially varying relationships; using maps of the local R^2^, it was possible to visualize how the relationships between the explanatory variables and the dependent variable vary across the study area. Considering that the COVID-19 pandemic is still ongoing, and more waves are being projected, the results obtained may be useful for local authorities in developing strategies to counter the pandemic. This is especially true for counties for which the GWR method has been able to identify deviations from the results obtained for the country as a whole. These counties can be regarded as a kind of anomaly, requiring specific, non-standard measures. Such a situation can be seen, for example, in Koszalin county, and Koszalin city county, Sławno county, and Szczecinek county (in the West Pomeranian voivodeship); Gorlice county in the Lesser Poland voivodeship; Jasło county in the Subcarpathian voivodeship; and Mława county, Ostrołęka county, Ostrów county, Płock county, Płock city county, Przasnysz country, Sierpc county, and Żuromin county (in the Masovian voivodeship), as well as the southwestern part of Poland, primarily Bolesławiec county, Lubań county, Lwówek county, Zgorzelec county, and Żagań county.

Similarly, the results of the local R^2^ for the variables indicating areas burdened with social problems resulting primarily from the economic exclusion of the population may be helpful to local authorities. The high association of the poverty problem with the spread of the COVID-19 pandemic has been shown above. Such a situation can be seen in particular, for example, in the counties in the central Masovian voivodeship: Białobrzegi county, Ciechanów county, Garwolin county, Grodzisk county, Grójec county, Kozienice county, Legionowo county, Warsaw city county, Warsaw West county, Maków Mazowiecki county, Mińsk Mazowiecki county, Nowy Dwór Mazowiecki county, Otwock county, Piaseczno county, Płońsk county, Pruszków county, Pułtusk county, Sochaczew county, Węgrów county, Wołomin county, Wyszków county, and Żyrardów county.

In-depth analysis combining the map of the SARS-CoV-2 cases, the maps of the indicator values, and the maps of the local R^2^ enabled the detection of counties where the local authorities should take specific measures, targeting the further fight against the COVID-19 pandemic in Poland. As an example, the following counties can be mentioned: Biała Podlaska county (Lubusz voivodeship), Koszalin county (West Pomeranian voivodeship), Warsaw city county, Kozienice county (Masovian voivodeship).

Biala Podlaska county is characterized by a negative R^2^ for the S2 variable “Physicians (total working staff) per 10,000 population”; from the “Social” section, there is a very high number of SARS-CoV-2 cases, while the number of working medical staff is considerably low. The present study indicates the need to increase the medical staff in this county in order to more effectively combat the COVID-19 pandemic. A similar situation (although to a lesser extent) is present in Koszalin county (West Pomeranian voivodeship).

In the “Environmental” section, many instances are noticeable where high levels of air pollution (represented by variables E1 and E2) largely explain the high number of cases of SARS-CoV-2. This is particularly evident in the counties of Warsaw city county, or Kozienice county (Masovian voivodeship). Warsaw, as the country’s capital, is particularly exposed to air pollution. Kozienice, the capital of Kozienice county, is the location of the second largest coal-fired power station in Poland. The largest coal-fired power station in Poland is located in Bełchatow county (Łódź voivodeship), which features distinctly in the air pollution maps, but the local R^2^ maps do not indicate a large influence of this phenomenon on the number of SARS-CoV-2 cases in this county; this may indicate that the local authorities are working effectively in this regard.

Such a thorough analysis of the set of maps prepared in this article can become a considerably useful tool for the authorities of a particular county, which could receive an answer to the question of to what extent to take action to further combat the COVID-19 pandemic. This might allow the specifying of actions in a particular direction in order to more effectively use the resources of the county or to confirm the effectiveness of the actions already being carried out. The proposed methodology is a ready-to-use tool for county authorities in Poland, but it can also be used in any other country with the same or greater level of spatial accuracy (lower level of administrative divisions). In this respect, this is an important innovative element of the article, as most of the previous studies focused only on the national level.

## Figures and Tables

**Figure 1 ijerph-19-11881-f001:**
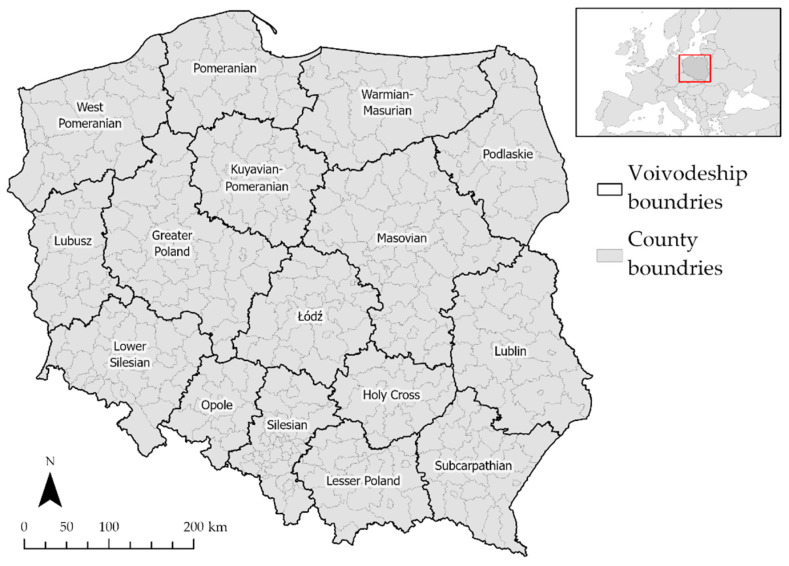
Study area. Source: own elaboration using ArcGIS Pro 2.9 by Esri.

**Figure 2 ijerph-19-11881-f002:**
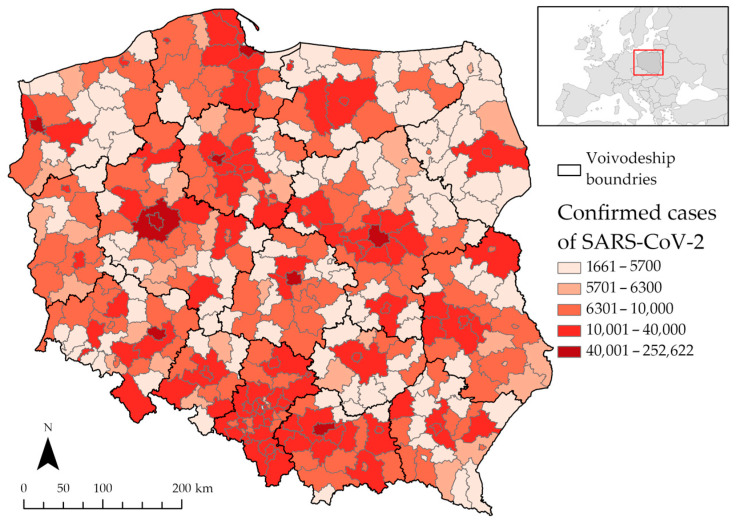
Summary of confirmed cases of SARS-CoV-2 in Polish counties. Source: own elaboration using ArcGIS Pro 2.9 by Esri.

**Figure 3 ijerph-19-11881-f003:**
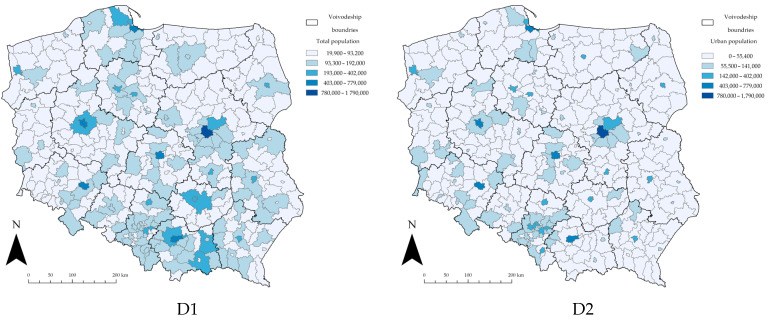
Spatial distribution of the explanatory variables in counties of Poland. Source: own elaboration using ArcGIS Pro 2.9 by Esri.

**Figure 4 ijerph-19-11881-f004:**
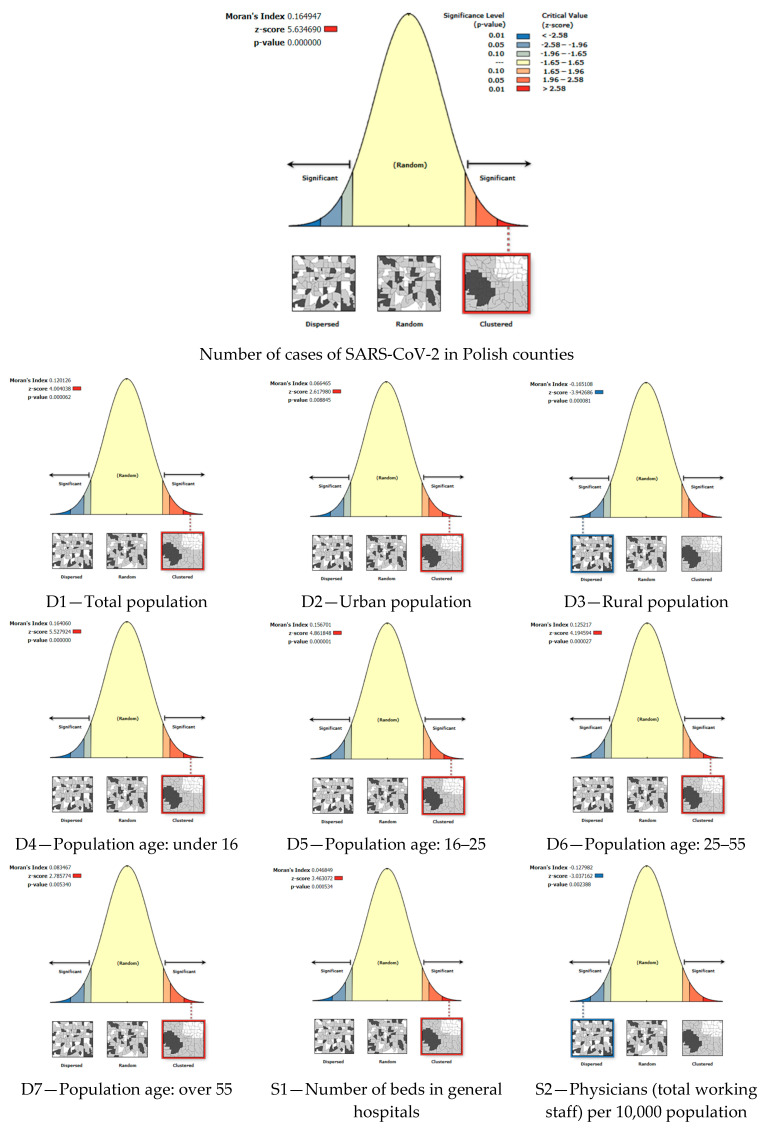
Spatial autocorrelation reports. Source: own elaboration using ArcGIS Pro 2.9 by Esri.

**Figure 5 ijerph-19-11881-f005:**
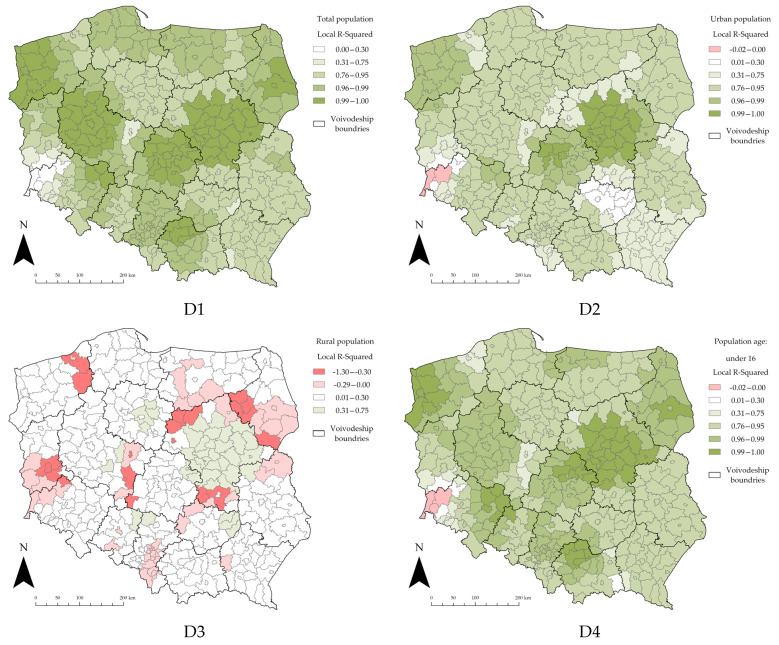
Local R^2^ estimates for the “Demographic” section. Source: own elaboration using ArcGIS Pro 2.9 by Esri.

**Figure 6 ijerph-19-11881-f006:**
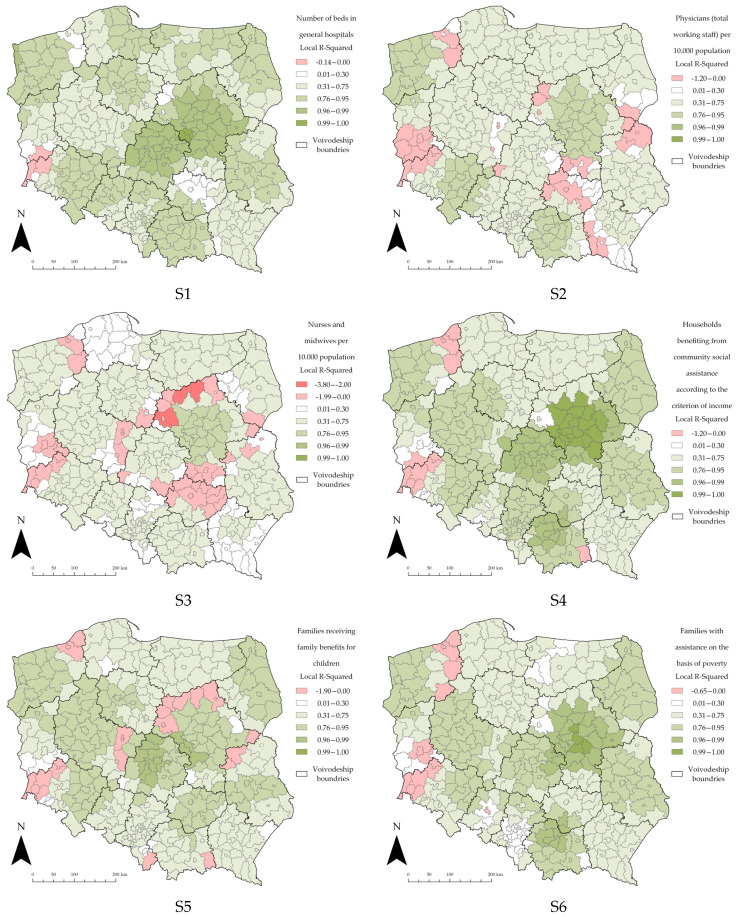
Local R^2^ estimates for the “Social” section. Source: own elaboration using ArcGIS Pro 2.9 by Esri.

**Figure 7 ijerph-19-11881-f007:**
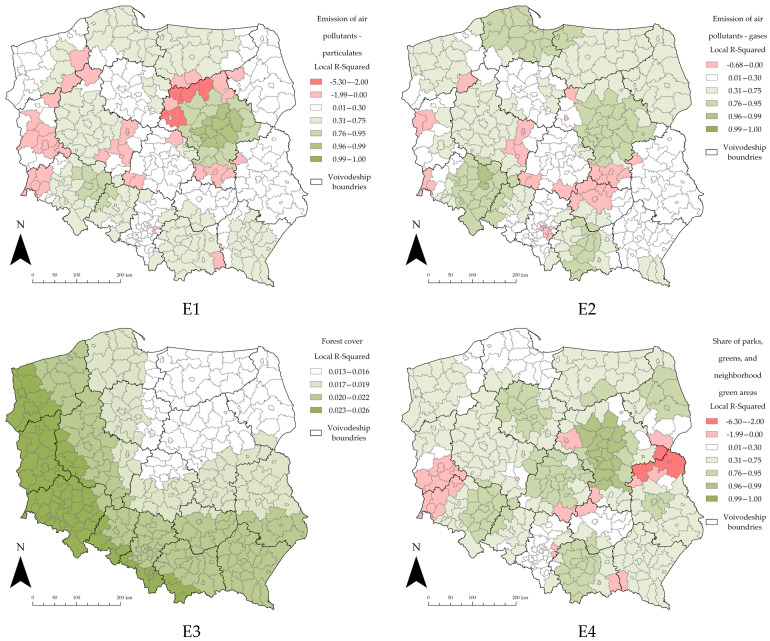
Local R^2^ estimates for the “Environmental” section. Source: own elaboration using ArcGIS Pro 2.9 by Esri.

**Table 1 ijerph-19-11881-t001:** Selected demographic, social, and environmental factors. Source: own elaboration on the basis of data from Statistics Poland.

Section	Factor (Variable)	Symbol
Demographic	Total population	D1
	Urban population	D2
	Rural population	D3
	Population age: under 16	D4
	Population age: 16–25	D5
	Population age: 25–55	D6
	Population age: over 55	D7
Social	Number of beds in general hospitals	S1
	Physicians (total working staff) per 10,000 population	S2
	Nurses and midwives per 10,000 population	S3
	Households benefiting from community social assistance according to the criterion of income	S4
	Families receiving family benefits for children	S5
	Families with assistance on the basis of poverty	S6
	Benefit payments from the 500+ program	S7
Environmental	Emission of air pollutants—particulates	E1
	Emission of air pollutants—gases	E2
	Forest cover	E3
	Share of parks, greens, and neighborhood green areas	E4

**Table 2 ijerph-19-11881-t002:** Summary of the spatial autocorrelation. Source: own elaboration on the basis of ArcGIS Pro 2.9 by Esri.

Variable Type	Variable	*p*-Value	Z-Score	Spatial Pattern	Confidence Level
Dependent	COVID-19 cases	0.00	5.63	Clustered	1%
Explanatory	D1—Total population	0.00	4.00	Clustered	1%
	D2—Urban population	0.01	2.62	Clustered	1%
	D3—Rural population	0.00	−3.94	Dispersed	1%
	D4—Population age: under 16	0.00	5.53	Clustered	1%
	D5—Population age: 16–25	0.00	4.86	Clustered	1%
	D6—Population age: 25–55	0.00	4.19	Clustered	1%
	D7—Population age: over 55	0.01	2.79	Clustered	1%
	S1—Number of beds in general hospitals	0.00	3.46	Clustered	1%
	S2—Physicians (total working staff) per 10,000 population	0.00	−3.04	Dispersed	1%
	S3—Nurses and midwives per 10,000 population	0.00	−4.67	Dispersed	1%
	S4—Households benefiting from community social assistance according to the criterion of income	0.00	2.95	Clustered	1%
	S5—Families receiving family benefits for children	0.00	6.30	Clustered	1%
	S6—Families with assistance on the basis of poverty	0.00	3.29	Clustered	1%
	S7—Benefit payments from the 500+ program	0.00	6.24	Clustered	1%
	E1—Emission of air pollutants—particulates	0.02	2.41	Clustered	5%
	E2—Emission of air pollutants—gases	0.03	2.24	Clustered	5%
	E3—Forest cover	0.00	10.41	Clustered	1%
	E4—Share of parks, greens, and neighborhood green areas	0.00	3.96	Clustered	1%

**Table 3 ijerph-19-11881-t003:** R^2^ and statistical description of the variables. Source: own elaboration on the basis of ArcGIS Pro 2.9 by Esri.

Variable	R^2^	Mean	Min	Max	STD
D1—Total population	0.99 **	101,006.8	19,914.0	1,790,658.0	119,730.5
D2—Urban population	0.95 **	60,613.3	0.0	1,790,658.0	122,692.5
D3—Rural population	0.37 **	40,393.4	0.0	264,014.0	35,060.6
D4—Population age: under 16	0.98 **	16,433.6	2855.0	301,697.0	19,744.3
D5—Population age: 16–25	0.97 **	9143.9	2054.0	112,725.0	8220.1
D6—Population age: 25–55	0.99 **	43,570.8	8382.0	797,514.0	53,035.6
D7—Population age: over 55	0.98 **	31,858.4	6623.0	578,722.0	39,298.0
S1—Number of beds in general hospitals	0.91 **	439.0	0.0	11,970.0	907.7
S2—Physicians (total working staff) per 10,000 population	0.73 **	41.1	2.0	204.9	30.4
S3—Nurses and midwives per 10,000 population	0.66 **	60.6	2.4	237.4	37.5
S4—Households benefiting from community social assistance according to the criterion of income	0.94 **	2171.1	472.0	20,186.0	1698.4
S5—Families receiving family benefits for children	0.86 **	2653.0	352.0	14,260.0	1739.9
S6—Families with assistance on the basis of poverty	0.91 **	1139.2	145.0	10,765.0	885.6
S7—Benefit payments from the 500+ program	0.97 **	80,276,961.1	15,183,262.0	1,366,004,134.0	90,144,295.6
E1—Emission of air pollutants—particulates	0.72 *	71.3	0.0	1924.0	143.3
E2—Emission of air pollutants—gases	0.70 *	522,212.5	0.0	32,882,772.0	2,138,844.5
E3—Forest cover	0.02 **	26.0	0.0	70.4	13.4
E4—Share of parks, greens, and neighborhood green areas	0.83 **	0.8	0.0	20.9	1.9

Note: **—statistically significant at the *p* < 0.01 level; *—statistically significant at the *p* < 0.05 level.

**Table 4 ijerph-19-11881-t004:** Spatial autocorrelation results for GWR residuals. Source: own elaboration on the basis of ArcGIS Pro 2.9 by Esri.

Variable	D1	D2	D3	D4	D5	D6	D7	S1	S2	S3	S4	S5	S6	S7	E1	E2	E3	E4
Residual z-score	−1.26	−1.28	0.66	−1.95	−1.44	−1.82	−1.93	−1.46	1.35	−0.81	−0.34	0.73	1.33	−1.7	0.75	0.33	0.38	0.01
Spatial pattern	R	R	R	wD	R	wD	wD	R	R	R	R	R	R	wD	R	R	R	R

Note: R—random; wD—weak dispersed.

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
