# Peer review of "Influence of the Demographic, Social, and Environmental Factors on the COVID-19 Pandemic—Analysis of the Local Variations Using Geographically Weighted Regression"

_ijerph, 2022, doi:10.3390/ijerph191911881_

Round 1

Reviewer 1 Report

With the method of GWR, the degree of explanation to the number of the SARS-CoV-2 cases with demographic, social, and environmental factors was explored, It shows that the demographic factor (namely the population density), has a great impact on COVID-19. The results may help understand the   significance for guiding the epidemic prevention and control.

Questions and Suggestions:
1. The number of cases of SARS-CoV-2 in counties of Poland used as the dependent variable was not appropriate. The incidence rate as the dependent variable is recommended, and the variable and influencing factors are best to be matched in the time.
2. There is less description of GWR method on infectious diseases in ref.
3. The reasons and principles for the selection of explanatory variables are not stated in the paper.
4. In the final discussion, excluding demographic factors, the average degree of social factors and environmental factors is relatively vague, please further elaborate, and clarify whether it is an important conclusion of the paper.
5. The paper innovation is insufficient, and the innovation points needs to be further summarized.

Author Response

We sincerely thank You for taking the time to review our article. Thank You for Your valuable suggestions to the manuscript, that helped us improve the paper and the approach of the study. We tried to address all of Your suggestions. Please see attached a step-by-step response to Your suggestions and comments. We hope that the improvements we have made will fully satisfy You and meet with Your positive evaluation.

Yours sincerely,

The Authors

Reviewer 2 Report

A very interesting paper with a subject actual and important.

I have some suggestions to improve the quality of the paper.

Abstract: could be improved in order to present more results and practical implications of the study.

The introduction is well-detailed. The authors presented the problem in a well way, showing the contribution. However, I suggest including the research question and the hypothesis of the study.

Concerning the data, I suggest including the range of the COVID-19 data in the manuscript (from march 2020 until the Month of the year). End 2021 means December of 2021?

I suggest including a Figure with all variables used in the manuscript. Moreover, I suggest including the data's descriptive statistics (Average, minimum, maximum, quartiles). I suggest including the description of variables.

Check the word “tabular” in line 155. What does means “tabular data”?

The null hypothesis established in lines 166-167 could be replaced by those used in Moran’s I. It is more suitable. It is just to change the paragraph.

Title of 3.1 could be replaced by Spatial Autocorrelation analysis. It is more suitable.

I suggest replacing the code for the name in Figure 3, Table 2. Also, replace the C/D in Table 2 for clustered and dispersed. You have space at the table.

In the discussion/conclusion, I suggest that the authors include practical implications of the results.

Author Response

(The authors gave the same response as above.)
